# Comparison of Short- and Long-Term Mortality in Patients with or without Cancer Admitted to the ICU for Septic Shock: A Retrospective Observational Study

**DOI:** 10.3390/cancers14133196

**Published:** 2022-06-29

**Authors:** Pierrick Le Borgne, Léa Feuillassier, Maleka Schenck, Jean-Etienne Herbrecht, Ralf Janssen-Langenstein, Celestine Simand, Justine Gantzer, Simon Nannini, Luc-Matthieu Fornecker, Karine Alamé, François Lefebvre, Vincent Castelain, Francis Schneider, Raphaël Clere-Jehl

**Affiliations:** 1Emergency Department, Hôpitaux Universitaires de Strasbourg, 67000 Strasbourg, France; lea.feuillassier@chru-strasbourg.fr (L.F.); karine.alame@chru-strasbourg.fr (K.A.); 2INSERM (French National Institute of Health and Medical Research), UMR 1260, Regenerative NanoMedicine (RNM), Fédération de Médecine Translationnelle (FMTS), University of Strasbourg, 67000 Strasbourg, France; 3Service de Médecine Intensive Réanimation, Hôpital de Hautepierre, Hôpitaux Universitaires de Strasbourg, 67000 Strasbourg, France; maleka.schenck@chru-strasbourg.fr (M.S.); jean-etienne.herbrecht@chru-strasbourg.fr (J.-E.H.); ralf.janssen-langenstein@chru-strasbourg.fr (R.J.-L.); vincent.castelain@chru-strasbourg.fr (V.C.); francis.schneider@chru-strasbourg.fr (F.S.); raphael.clere-jehl@chru-strasbourg.fr (R.C.-J.); 4Institut de Cancérologie Strasbourg Europe, Service d’Hématologie, 67200 Strasbourg, France; c.simand@icans.eu (C.S.); lm.fornecker@icans.eu (L.-M.F.); 5Institut de Cancérologie Strasbourg Europe, Service d’Oncologie Médicale, 67200 Strasbourg, France; j.gantzer@icans.eu (J.G.); s.nannini@icans.eu (S.N.); 6Department of Public Health, University Hospital of Strasbourg, 67000 Strasbourg, France; francois.lefebvre@chru-strasbourg.fr; 7INSERM (French National Institute of Health and Medical Research), UMR_S1109, Immuno-Rhumatologie Moléculaire, Centre de Recherche d’Immunologie et d’Hématologie, Fédération de Médecine Translationnelle (FMTS), University of Strasbourg, 67000 Strasbourg, France

**Keywords:** septic shock, critical care, neoplasms, immunocompromised host, treatment outcome

## Abstract

**Simple Summary:**

Patients with malignancies (solid tumors or hematological diseases) are at high risk of developing sepsis and are increasingly admitted to ICU due to their long-term prognosis improvement. The main objective of this study was to compare the short- and long-term prognoses of patients with and without cancer admitted to the ICU for septic shock. In-hospital and ICU mortality, as well as LOS, were not different in SSh patients with and without cancer, suggesting that malignancies should no longer be considered a barrier to ICU admission.

**Abstract:**

Introduction: Cancer patients are at high risk of developing septic shock (SSh) and are increasingly admitted to ICU given their improved long-term prognosis. We, therefore, compared the prognosis of cancer and non-cancer patients with SSh. Methods: We conducted a monocentric, retrospective cohort study (2013–2019) on patients admitted to ICU for SSh. We compared the clinical characteristics and management and studied short- and long-term mortality with ICU and in-hospital mortality and 1-year survival according to cancer status. Results: We analyzed 239 ICU stays in 210 patients, 59.5% of whom were men (n = 125), with a median age of 66.5 (IQR 56.3–77.0). Of the 121 cancer patients (57.6% of all patients), 70 had solid tumors (33.3%), and 51 had hematological malignancies (24.3%). When comparing ICU stays of patients with versus without cancer (n = 148 vs. n = 91 stays, respectively), mortality reached 30.4% (n = 45) vs. 30.0% (n = 27) in the ICU (*p* = 0.95), and 41.6% (n = 59) vs. 35.6% (n = 32) in hospital (*p* = 0.36), respectively. ICU length of stay (LOS) was 5.0 (2.0–11.3) vs. 6.0 (3.0–15.0) days (*p* = 0.27), whereas in-hospital LOS was 25.5 (13.8–42.0) vs. 19.5 (10.8–41.0) days (*p* = 0.33). Upon multivariate analysis, renal replacement therapy (OR = 2.29, CI95%: 1.06–4.93, *p* = 0.03), disseminated intravascular coagulation (OR = 5.89, CI95%: 2.49–13.92, *p* < 0.01), and mechanical ventilation (OR = 7.85, CI95%: 2.90–21.20, *p* < *0.01*) were associated with ICU mortality, whereas malignancy, hematological, or solid tumors were not (OR = 1.41, CI95%: 0.65–3.04; *p* = 0.38). Similarly, overall cancer status was not associated with in-hospital mortality (OR = 1.99, CI95%: 0.98–4.03, *p* = 0.06); however, solid cancers were associated with increased in-hospital mortality (OR = 2.52, CI95%: 1.12–5.67, *p* = 0.03). Lastly, mortality was not significantly different at 365-day follow-up between patients with and without cancer. Conclusions: In-hospital and ICU mortality, as well as LOS, were not different in SSh patients with and without cancer, suggesting that malignancies should no longer be considered a barrier to ICU admission.

## 1. Introduction

Sepsis is due to an imbalanced immune response of the host to an infection, resulting in a life-threatening organ dysfunction [1]. This dysregulated immune response is initially dominated by a life-threatening hyperinflammatory state, which is involved in the occurrence of organ failure [2]. The importance of the immune response in this pathophysiology highlights that sepsis is primarily a host-related disease [3]. Some patients, such as cancer patients, are more likely to develop sepsis and septic shock (SSh) due to their underlying immune impairment, which may result from the malignancy itself, or from cancer therapies [4,5,6]. According to recent data, more than one in five sepsis hospitalizations are cancer-related [7]. Among cancer patients, one year after malignancy diagnosis, the incidence of sepsis is around 3.7% and it accounts for approximately 5% of hospitalizations (all causes combined) [4,8].

In addition to the high frequency of sepsis in cancer patients, they display particularities in terms of prognosis. Recent studies still report higher in-hospital mortality in patients with cancer-related sepsis, around 28%, versus 19.5% in non-cancer-related sepsis [7]. Although death may be attributable to sepsis or to the underlying malignancy, infection remains among the top non-cancer causes of death among hematology–oncology patients [9,10]. Moreover, the type of infection varies from immunocompetent to cancer patients, including the involvement of more opportunistic microorganisms [11]. Additionally, although post-chemotherapy neutropenia remains a frequent mechanism of immunosuppression, novel anticancer treatments induce less profound neutropenia [12]. Therefore, the prognosis of cancer patients with sepsis is most likely to change over time, underlining the importance of updated assessments of practice in this field.

Despite available data on cancer patients with sepsis, the literature is scarcely focused on cancer patients with septic shock in the intensive care unit (ICU), which is the most severe form of sepsis. Septic shock is defined, in non-cancer patients, by the association of sepsis with metabolic abnormalities, translated by arterial blood lactate above 2 mmol/L, with persistent hypotension despite fluid therapy and a vasopressor requirement to maintain a mean arterial pressure at 65 mmHg or above [1,12]. Although some data exist regarding short-term mortality in cancer patients with sepsis, long-term survival is partly unknown. Attention should particularly be paid to late mortality, which might be due to the underlying malignancy or to the occurrence of complications due to immunodeficiency in addition to other late-sepsis complications that occur even in immunocompetent patients [12]. In short, knowledge of long-term survival remains crucial to assessing the appropriateness of ICU admission.

Overall, patients with malignancies (solid tumors or hematological diseases) are at high risk of developing sepsis and are increasingly admitted to ICU due to their long-term prognosis improvement [12,13,14,15]. Therefore, we aimed to study septic shock in a population of ICU patients with malignancies, in terms of clinical characteristics, prognosis, and treatment. The main objective of this study was to compare the short- and long-term prognoses of patients with and without cancer admitted to the ICU for septic shock.

## 2. Methods

### 2.1. Study Population and Settings

This retrospective and monocentric cohort study analyzed all patients admitted to the ICU for septic shock (recorded as the main diagnosis by the Medical Information and statistics Department according to the definition of the International Classification of Diseases, ICD-10) over a 7-year period (1 January 2013–31 December 2019). The study was conducted in a medical ICU of a university hospital. This department receives an average of 950 patients per year, with a mean SAPS II of 50 points, an average length of stay of 9 days, and an average mortality rate of 19.8% over the study period. Cancer patients are admitted to this ICU in the event of organ failure at a rate of approximately 80 admissions/year, due to its proximity to the regional reference center for solid tumors and hematological diseases. Patients with coding errors were excluded. Additionally, if the same patient had multiple admissions to the ICU for septic shock over the study period, we included each stay independently. Ultimately, two distinct groups were identified: septic shock in non-cancer patients and septic shock in patients with solid tumors or hematological diseases.

### 2.2. Data Collection

We retrospectively compiled data from patients’ electronic medical records and then standardized it in a report file. The collected data included epidemiological, clinical, and biochemical parameters. The main epidemiological characteristics such as age, sex, and weight were reported. The heaviness of comorbidities was measured by the Charlson Comorbidity Index [16]. Autonomy was measured by the Knaus Chronic health status score [17]. We detailed therapies implemented before admission to the ICU (fluid resuscitation, vasopressor support, and antibiotic therapy) and those implemented during the stay in the ICU (mechanical ventilation, antibiotic therapy, renal replacement therapy, and transfusion support). The cause of septic shock was documented whenever possible. Additionally, we recorded the presence of bacteremia, neutropenia, the duration of antibiotic therapy, and blood transfusion strategies. Severity at ICU admission was measured by the SAPS II score [18]. Lastly, we studied the entire hospital stay, including the ICU stay and the subsequent stay in the ward, which allowed us to have a global view of our study population in terms of length of stay, 28-day mortality, in-hospital mortality, and actuarial survival over 1 year.

### 2.3. Ethics

This study was approved by our local academic ethics committee (reference CE: 2020-1), which, in accordance with the French legislation, waived the need for informed consent.

### 2.4. Statistical Analysis

The descriptive statistical analysis of the categorical variables was performed by giving the frequencies and the proportions of each value. For each continuous variable, the median and the first and third quartiles were given. Two-group comparisons of continuous covariates were performed by Mann–Whitney U test. Comparisons between categorical variables were determined using the chi-squared test or Fisher’s exact test in case of expected values below 5 in any of the cells of a contingency table. Then, two multivariate logistic models were obtained with the statistically significant and clinically relevant covariates—one for ICU mortality and one for in-hospital mortality. For survival analysis, time-to-event curves were generated with the use of the Kaplan–Meier method, and the groups were compared using the log-rank test. Analyses were performed with R 4.0.2 software in its most up-to-date version at the time of the analysis, including all the software packages required to carry out the analysis.

## 3. Results

### 3.1. Baseline Characteristics of the Study Population

Over the 7-year study period, 249 ICU stays were recorded as septic shock (SSh). After excluding 10 patients presenting with an alternative diagnosis, a total of 239 stays were finally included (210 patients) (Figure 1). The 210 included patients were composed of 89 non-cancer patients (42.4%) and 121 cancer patients (57.6%), and the latter was stratified into 70 with solid tumors (33.3%) and 51 with hematological malignancies (24.3%), as detailed in Appendix A. The 89 non-cancer patients and the 121 cancer patients were, respectively, involved in 91 and 148 hospital and ICU stays.

The baseline characteristics of the study population are summarized in Table 1. The median age of these patients was 66.5 years (IQR 56.3–77.0), and 125 were male (59.5%). Comorbidities were mostly of cardiovascular origin, and more than half of the study population had a history of hypertension (53.1%; n = 111). Almost 60% had a history of prior ICU stay (n = 125). Among the 239 stays for septic shock, prior to ICU admission, fluid resuscitation was initiated in 76.1% (n = 178) and antibiotics were administered in 77.4% (n = 185), and norepinephrine was initiated in 33.9% (n = 80) of stays (Table 2). In the ICU, renal replacement therapy was implemented in 29.3% (n = 70) of stays, and mechanical ventilation was required in 59.8% (n = 143) of stays for a median duration of 5 days [2,3,4,5,6,7,8,9,10,11]. In 61.9% of stays (n = 154) with positive microbiological findings (Appendix A), Gram-negative bacilli were the most frequent, accounting for 38.1% (n = 91).

### 3.2. Comparisons between Non-Cancer Patients versus Cancer Patients with SSh

When comparing SSh patients with and without cancer in univariate analysis (Table 1), cancer patients were younger (64.0 years [56.0–75.0] vs. 69.0 [57.0–82.0], *p* = 0.02), had a lower body mass index (24.2 kg/m^2^ [21.3–27.7] vs. 26.1 [23.3–30.1]; *p* < 0.01), along with less diabetes mellitus (23.3% (n = 28) vs. 38.2% [n = 34]; *p* = 0.02) and cognitive impairment (3.3% (n = 4) vs. 22.5% (n = 20); *p* < 0.01), but had a more frequent history of prior ICU stay (68.6% (n = 83) vs. 47.2% (n = 42), *p* < 0.01). During their ICU stays (Table 2), management differed by the introduction of antifungal therapy (59.5% (n = 88) vs. 36.3% (n = 33); *p* < 0.01), along with increased blood transfusions (72.3% (n = 107) vs. 49.5% (n = 45); *p* < 0.01) in cancer patients. Notably, neither lengths of stay nor mortality differed between the two groups, in the ICU or in hospital. Thus, mortality reached 30.4% (n = 45) vs. 30.0% (n = 27) in the ICU (*p* = 0.95), and 41.6% (n = 59) vs. 35.6% (n = 32) in hospital (*p* = 0.36), in cancer patients vs. non-cancer patients, respectively. ICU length of stay (LOS) was 5.0 (2.0–11.3) vs. 6.0 (3.0–15.0) days (*p* = 0.27), whereas in-hospital LOS was 25.5 (13.8–42.0) vs. 19.5 (10.8–41.0) days (*p* = 0.33), respectively. In multivariate analysis (Appendix A), only the age (OR = 0.93, CI95%: 0.90–0.97; *p* < 0.01) and blood transfusions (OR = 4.86, CI95%: 1.47–16.04, *p* < 0.01) were independently associated with cancer status. Mortality did not significantly differ in the ICU (OR = 1.55, CI95%: 0.45–5.31; *p* = 0.49) or in hospital (OR = 1.48, CI95%: 0.45–4.83; *p* = 0.52), as was the case for the LOS.

### 3.3. Multivariate Analysis of Factors Associated with Mortality

Based on multivariate analysis (Table 3), only renal replacement therapy (OR = 2.29, CI95%: 1.06–4.93; *p* = 0.03), DIC (OR = 5.89, CI95%: 2.49–13.92; *p* < 0.01) and mechanical ventilation (OR = 7.85, CI95%: 2.90–21.20, *p* < 0.01) were independently associated with ICU mortality, unlike cancer status, which was not (OR = 1.41, CI95%: 0.65–3.04; *p* = 0.38). In parallel, cancer status (OR = 1.99, CI95%: 0.98–4.03, *p* = 0.06) and hematological malignancies (OR = 1.62, CI95%: 0.67–3.87, *p* = 0.28) were not associated with in-hospital mortality. However, solid tumors were associated with in-hospital mortality (OR = 2.52, CI95%: 1.12–5.67, *p* = 0.03).

### 3.4. Long-Term Survival Outcome

A one-year follow-up was obtained for 98.3% (n = 119) of the 121 cancer patients and for 94.4% (n = 84) of the 89 non-cancer patients. After analyzing Kaplan–Meier survival curves (Figure 2), it was revealed that mortality at 365 days reached 58.0% (n = 69/119) among cancer patients versus 47.6% (n = 40/84) in non-cancer patients, without statistical significance according to the log-rank test (*p* = 0.53).

### 3.5. Medical Economics Analysis

Medical economic data were analyzed for both groups, during their ICU stay and over their entire hospital stay (Appendix A). There was no significant difference between non-cancer and cancer patients admitted for SSh in terms of social security retribution per ICU stay (EUR 7718 IQR: [2752–15,053] vs. EUR 6742 IQR: [2463–14,831], respectively; *p* = 0.67). Unlike the cost of hospital stays, which was higher in cancer patients (EUR 12,020 IQR: [7902–24,294] vs. EUR 18,802 IQR: [9389–32,551], respectively; *p* = 0.01).

## 4. Discussion

This study is one of the main studies focusing on cancer patients admitted to ICU for septic shock, which is the most severe form of sepsis. Among the main results, we note the slight mortality rate differences between cancer versus non-cancer patients.

Previous studies addressing sepsis survival in oncology patients display conflicting results, mainly depending on the study period and patient recruitment policy. Sepsis mortality has declined over the past three decades in cancer patients, with in-hospital mortality dropping from 50–60% in the 1990s and early 2000s [19,20,21,22] to around 30% in the 2010s [7,10,23]. Interestingly, there is a trend toward a reduction in sepsis-induced mortality differences between patients with and without malignancies in the same period. Thus, the hospital mortality among cancer patients with sepsis was 52–55% higher than that among non-cancer patients in the 1990s [4,24]. This difference in mortality rates decreased over time [7,25,26,27,28], albeit with a difference between solid and hematological malignancies, the latter having the highest incidence of sepsis [10] and the closest sepsis survival compared with those of non-cancer patients [23,25,26,27].

A similar evolution is supported by a few recent studies focusing on septic shock—similarly to ours—in oncology patients, in which 28-day and in-hospital mortality were about 48% and 52%, respectively [25,26]. Likewise, septic shock patients with hematologic malignancies displayed a lower 28-day and in-hospital mortality of 39% and 44%, respectively [25,26]. Therefore, the in-hospital mortality of 42% in our cancer patient cohort is consistent with the recent literature data. Interestingly, this mortality rate is very close to non-cancer patients. Indeed, based on a recent extensive meta-analysis, hospital mortality in septic shock patients, regardless of cancer comorbidities, was estimated at 39% [29].

Our high proportion of hematological patients might explain some part of this relatively low mortality rate. The multivariate analysis shows that solid tumors are independently associated with in-hospital mortality in our study (OR 2.7 [1.2–5.9]), unlike hematological malignancies (OR 1.7 [0.7–4.1]). This result in hematological patients may result from a lack of statistical power, due to our relatively small number of patients, which constitutes one of the main limits of the study. However, the “hematological” and “solid tumor” subgroups, each composed of more than 50 patients, support the hypothesis of either a better prognosis in hematological patients or a selection of patients for the ICU. Although the number of hematological patients in our study was about one-third less than that in other major recent studies dealing specifically with septic shock in cancer patients, these previous studies also highlighted the absence of a significant difference in mortality between the subgroups of hematological versus non-oncology patients, even after multivariate analyses [25,26]. This contrasts with the poor prognosis attributed to hematological patients presenting with sepsis until the early 2000s [30] and can be linked to the remarkable therapeutic progress in several hematological diseases in the last two decades [31], among the best examples of which are lymphomas [32], well-represented in studies addressing sepsis in hematological patients [23,25]. Notably, a better knowledge of malignant diseases may have contributed to targeting patients with a better cancer prognosis. Therefore, these results are probably obtained in previously selected patients, in whom the underlying malignancy infrequently influences short-term survival following admission to the ICU [33]. Indeed, our high one-year survival rate indicates that most patients seem to have achieved remission of their malignant disease, at the time of the ICU stay or thereafter. Regarding solid tumors, due to the organization of our university hospital, this monocentric study population contains very few patients with lung cancer, which is an independent factor of poor prognosis [4]. This may have contributed to the overall outcome of our patients.

Notably, we studied long-term follow-up, while data beyond 90 days are very scarce in former studies. We considered the importance of a minimal follow-up of 365 days, due to the growing importance of investigating long-term prognosis. Only 3.3% of patients were lost to follow-up, lending credibility to our results, despite the retrospective nature of the study, which is also a limitation. Our 180-day mortality rate was 51% in cancer patients, which is close to the results reported by Camou et al. [25] and highlights the long-term prognosis improvement.

Regarding our study population, cancer and non-cancer patients differed at baseline by age (4 years younger in cancer patients) and comorbidities. Additionally, non-cancer patients were less likely to be overweight, cognitively impaired, and suffer increased chronic kidney and respiratory diseases. In multivariate analysis, these differences were considered. In previous studies, cancer patients also displayed younger age and fewer comorbidities [25,26], confirming that oncology patients were generally selected before ICU admission.

The lung and the gastrointestinal tract were the main sites of infection, as also indicated in almost all previous studies [20,22,25,26,27]. We found a similar SAPS II between non-cancer and cancer patients, leveling at 56 points, as in the previous series including patients with septic shock [21,25]. It may be noted that our study period started in 2013, i.e., 3 years before the definition criteria of *Sepsis-3* [1], which explains why roughly 5% of patients were diagnosed with septic shock without receiving norepinephrine [34], in both groups.

Cancer patients received more blood transfusions, as expected, in uni- and multivariate analyses, related to the significantly more marked cytopenia in this group. Interestingly, disseminated intravascular coagulation was independently associated with sepsis mortality, as in previous studies [35,36], but its incidence was similar in both groups, about 19%, which implies a central origin for the more frequent thrombocytopenia in cancer patients. In the same way, neutropenia was more common in cancer patients, in whom it is presumed to be chemotherapy-induced or, less frequently, the result of bone marrow invasion [37,38]. It must be emphasized that, in this study, consistent with previous studies, neutropenia was not independently associated with mortality [39].

Additionally, in concordance with previous studies dealing with sepsis in cancer patients, mechanical ventilation and renal replacement therapy were independently associated with mortality [23,25,26,28]. However, mechanical ventilation and renal replacement therapy are increasingly used in cancer patients with sepsis since the early 2000s, and our study reinforces the fact that the mortality of patients with organ failures, including patients requiring mechanical ventilation, has decreased over time [23,28]. As an example, mechanical ventilation was strongly associated with mortality, with the odds ratio reaching 20 in the early 2000s [30] versus 4.7 in our study and other recent studies [23].

Finally, the length of ICU stay of cancer patients admitted for septic shock was similar to those of non-cancer patients, as was the cost of hospital stay. Although the organization of health care systems is different depending on the country, this point is a major change compared with previous scant medico-economic data [4] and is likely to be the result of improvements in patient care and outcome [40]. This last argument should bring forward the reluctance linked to the cost-effectiveness ratio in admitting cancer patients to ICU.

### Limitations

Among the limitations of this study, the retrospective nature of the study did not allow us to include the role of specific species of infectious agents (among bacteria or fungi) in the analysis of the outcome, nor was it possible to assess the impact of front-line appropriate (or not) antimicrobial therapy for neutropenic fever. It was also not possible for us to consider the outcome analysis of the complete remission status of the underlying malignancy or the cytotoxic agent regimen schedules used. It is worth noting that this study included very few patients with severe post-chemotherapy neutropenia.

## 5. Conclusions

Altogether, by focusing specifically on cancer patients with septic shock, this study complements the recent literature and confirms the improvements in short- and long-term prognoses of those patients. Mortality attributable to sepsis, septic shock, and underlying malignancies decreased over the last two decades. Although a global medical assessment remains essential in the therapeutic decision, promoting access of cancer patients with sepsis to ICU is a useful contribution to improving their long-term prognosis. Moreover, these patients should no longer be excluded from clinical trials on septic shock in the ICU.

## Figures and Tables

**Figure 1 cancers-14-03196-f001:**
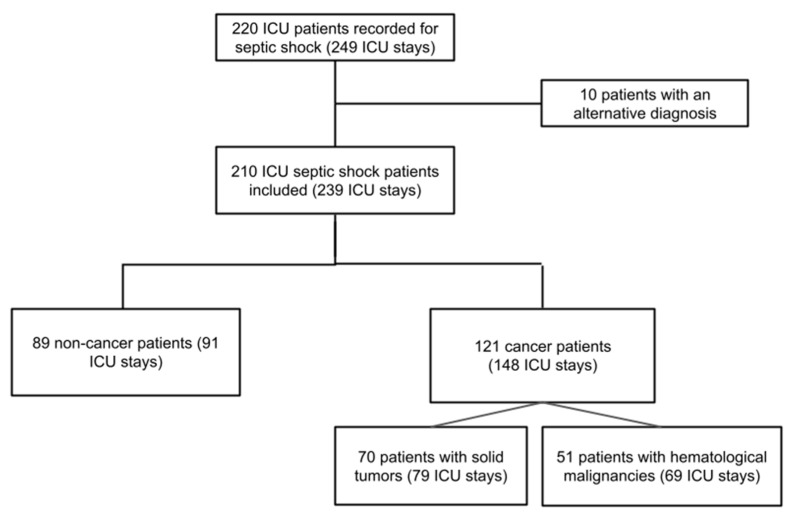
Flowchart of the study population. Abbreviations: ICU, intensive care unit.

**Figure 2 cancers-14-03196-f002:**
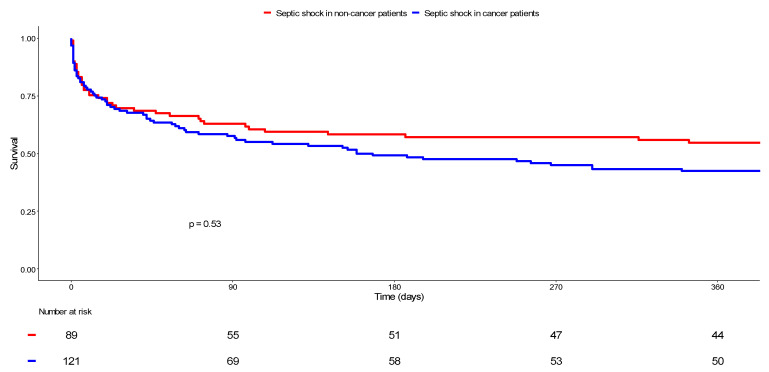
Comparison of survival between cancer and non-cancer patients with septic shock**.** Kaplan–Meier curves display the survival of non-cancer patients with septic shock (in red) and cancer patients with septic shock (in blue). The resulting *p* value for the log-rank test was 0.53.

**Table 1 cancers-14-03196-t001:** Baseline characteristics of septic shock patients.

	All Patients N = 210	Non-Cancer Patients n = 89	Cancer Patients n = 121	*p* Value
Age (years)	66.5 [56.3–77.0]	69.0 [57.0–82.0]	64.0 [56.0–75.0]	**0.02 ***
Male	125 (59.5)	52 (63.4)	73 (57.0)	0.36
**Comorbidities**				
Hypertension	111 (53.1)	54 (60.7)	57 (47.5)	0.06
Body mass index	25.1 [21.9–29.1]	26.1 [23.3–30.1]	24.2 [21.3–27.7]	**<0.01 ***
Cardiovascular diseases	88 (41.9)	44 (49.4)	44 (36.4)	0.06
Diabetes mellitus	62 (29.7)	34 (38.2)	28 (23.3)	**0.02 ***
Pre-existing renal failure	28 (13.3)	16 (18.0)	12 (9.9)	0.09
Respiratory diseases	68 (32.4)	35 (39.3)	33 (27.3)	0.07
Knaus score	6.0 [5.0–6.0]	6.0 [5.0–6.0]	6.0 [6.0–6.0]	**<0.01 ***
Charlson Comorbidity Index	6.0 [3.0–9.0]	5.0 [2.0–7.0]	7.0 [4.0–10.0]	**<0.01 ***
Cognitive impairment	24 (11.4)	20 (22.5)	4 (3.3)	**<0.01 ***
History of prior ICU stay	125 (59.5)	42 (47.2)	83 (68.6)	**<0.01 ***
**Long-term mortality** ^#^				
*At d90*	*84 (40.4)*	*33(37.5)*	*51 (42.5)*	*0.47*
*At d180*	*98 (47.3)*	*37 (42.0)*	*61(51.3)*	*0.19*
*At d365 (1 year)*	*109 (53.7)*	*40 (47.6)*	*69 (58.0)*	*0.14*

Footnotes: Data are all expressed in median IQR [25–75] or n/N (%) where n/N is the total number of patients with available data. ^#^ Mortality was assessed from the day of the first admission to ICU for septic shock if the patient has had several stays. * *p* < 0.05. Abbreviations: d90, day 90 of hospitalization; ICU, intensive care unit.

**Table 2 cancers-14-03196-t002:** Characteristics of the ICU stay for septic shock according to cancer status.

	All Stays N = 239	Stays of Non-Cancer Patients n = 91	Stays of Cancer Patients n = 148	*p* Value
**Treatments before ICU admission**				
Fluid therapy	178 (76.1)	67 (74.4)	111 (77.1)	0.65
Norepinephrine	80 (33.9)	32 (35.2)	48 (33.1)	0.74
**Site of infection**				
Respiratory tract	104 (43.5)	42 (46.2)	62 (41.9)	0.52
Urinary tract	58 (24.3)	25 (27.5)	33 (22.3)	0.36
Gastrointestinal tract	75 (31.4)	28 (30.8)	47 (31.8)	0.87
Skin and soft tissue	27 (11.3)	18 (19.8)	9 (6.1)	**<0.01 ***
CRBSI	19 (8.0)	3 (3.3)	16 (10.8)	**0.04 ***
Central nervous system	1 (0.4)	1 (1.1)	0 (0.0)	0.76
Osteoarticular system	7 (2.9)	6 (6.6)	1 (0.7)	**0.03 ***
Other sites	10 (4.2)	7 (7.7)	3 (2.0)	0.08
Unknown site	23 (9.6)	3 (3.3)	20 (13.5)	**<0.01 ***
**Laboratory findings ^#^**				
Creatinine (μmol/L)	130.0 [87.1–192.7]	134.0 [92.0–236.0]	118.0 [83.4–172.3]	**0.03 ***
CRP (mg/L)	166.5 [92.7–249.5]	172.5 [99.1–249.1]	163.0 [84.0–249.3]	0.42
Lactate (mmol/L)	2.6 [1.7–4.1]	2.7 [1.8–4.1]	2.4 [1.6–4.2]	0.98
Hemoglobin (g/dL)	9.6 [8.2–11.0]	10.5 [8.9–12.3]	9.1 [7.8–10.2]	**<0.01 ***
WBC count (10^9^/L)	14.1 [9.1–22.3]	16.4 [11.5–23.1]	12.3 [5.3–21.6]	**<0.01 ***
*Neutrophil count* *(10^9^/*L*)*	*11.7 [6.6–19.0]*	*14.0 [10.1–20.2]*	*9.5 [3.8–18.4]*	**<*0.01 ****
Platelet count (10^9^/L)	133.0 [63.0–235.0]	177.0 [110.5–270.5]	105.0 [34.0–211.5]	**<0.01 ***
**ICU stay**				
SAPS II	51.0 [42.0–68.0]	50.0 [43.5–65.5]	55.5 [41.8–68.0]	0.44
Duration of ATB therapy (days)	5.0 [3.0–10.5]	6.0 [3.0–11.0]	5.0 [2.0–10.3]	0.25
Antifungal therapy	121 (50.6)	33 (36.3)	88 (59.5)	**<0.01 ***
Catecholamines	226 (94.6)	87 (95.6)	139 (93.9)	0.81
Renal replacement therapy	70 (29.3)	33 (36.3)	37 (25.0)	0.06
NIV	55 (23.0)	26 (28.6)	29 (19.6)	0.11
Mechanical ventilation	143 (59.8)	60 (65.9)	83 (56.1)	0.13
*Duration of MV* (days)	*5.0 [2.0–11.0]*	*6.0 [2.8–15.5]*	*5.0 [2.0–10.5]*	*0.38*
DIC	44 (18.6)	18 (20.0)	26 (17.8)	0.67
MODS	79 (33.1)	36 (39.6)	43 (29.1)	0.09
ARDS	26 (10.9)	9 (9.9)	17 (11.5)	0.70
Transfusions	152 (63.6)	45 (49.5)	107 (72.3)	**<0.01 ***
*Red blood cells*	*143 (59.8)*	*43 (47.3)*	*100 (67.6)*	<***0.01 ****
*Fresh frozen plasma*	*51 (21.3)*	*18 (19.8)*	*33 (22.3)*	*0.64*
*Platelet concentrates*	*87 (36.4)*	*18 (19.8)*	*69 (46.6)*	<***0.01 ****
Advance directives	13 (5.4)	6 (6.6)	7 (4.7)	0.73
LTE in the ICU	47 (19.7)	18 (19.8)	29 (19.6)	0.97
**Outcome**				
ICU LOS (days)	5.0 [3.0–13.0]	6.0 [3.0–15.0]	5.0 [2.0–11.3]	0.27
Total LOS (days)	24.0 [12.0–42.0]	19.5 [10.8–41.0]	25.5 [13.8–42.0]	0.33
**Mortality**				
*At d28*	*65 (27.3)*	*27 (29.7)*	*38 (25.9)*	*0.52*
*In the ICU*	*72 (30.3)*	*27 (30.0)*	*45 (30.4)*	*0.95*
*In hospital*	*91 (39.2)*	*32 (35.6)*	*59 (41.6)*	*0.36*

Footnotes: Data are all expressed in median IQR [25–75] or n/N (%) where n/N is the total number of patients with available data. ^#^ Blood tests on ICU admission. * *p* < 0.05. Abbreviations: ARDS, acute respiratory distress syndrome; ATB, antibiotics; CRBSI, catheter-related bloodstream infection; CRP, C-reactive protein; d28, day 28 of hospitalization; DIC, disseminated intravascular coagulation; ICU, intensive care unit; LOS, length of stay; LTE, limitation of therapeutic effort; MODS, multiple organ dysfunction syndrome; MV, mechanical ventilation; NIV, non-invasive ventilation; SAPS II, simplified acute physiology score II; SSh, septic shock; WBC, white blood cells.

**Table 3 cancers-14-03196-t003:** Multivariate analysis of factors associated with (**a**) ICU mortality and (**b**) in-hospital mortality.

**(a)** ** ** **ICU Mortality**	**Odds Ratio**	**95%CI**	***p* Value**
Age (years)	1.00	[0.98–1.02]	0.91
Cancer patients with SSh	1.41	[0.65–3.04]	0.38
*Solid tumors*	*1.48*	*[0.62–3.51]*	*0.37*
*Hematological malignancies*	*1.33*	*[0.53–3.32]*	*0.54*
ARDS	1.03	[0.39–2.75]	0.95
Renal replacement therapy	2.29	[1.06–4.93]	**0.03 ***
DIC	5.89	[2.49–13.92]	**<0.01 ***
Mechanical ventilation	7.85	[2.90–21.20]	**<0.01 ***
Nosocomial infection	1.82	[0.89–3.69]	0.10
Bacteremia	1.39	[0.67–2.88]	0.37
**(b)** ** In-hospital mortality**			
Age (years)	1.01	[0.99–1.04]	0.19
Cancer patients with SSh	1.99	[0.98–4.03]	0.06
*Solid tumors*	*2.52*	*[1.12–5.67]*	* **0.03 *** *
*Hematological malignancies*	*1.62*	*[0.67–3.87]*	*0.28*
ARDS	1.24	[0.47–3.26]	0.66
Renal replacement therapy	2.50	[1.20–5.20]	**0.01 ***
DIC	3.70	[1.62–8.46]	**<0.01 ***
Mechanical ventilation	4.56	[2.15–9.65]	**<0.01 ***
Nosocomial infection	1.73	[0.91–3.29]	0.10
Bacteremia	1.30	[0.68–2.50]	0.42

Footnotes: * *p* < 0.05. Abbreviations: ARDS, acute respiratory distress syndrome; DIC, disseminated intravascular coagulation; SSh, septic shock.

## Data Availability

All data analyzed as part of the study are included.

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
