# Peer review of "Comparison of Short- and Long-Term Mortality in Patients with or without Cancer Admitted to the ICU for Septic Shock: A Retrospective Observational Study"

_cancers, 2022, doi:10.3390/cancers14133196_

Round 1
Reviewer 1 Report
Thank you for giving me the opportunity to review this manuscript. Authors compared the prognosis of patients admitted to the ICU with septic shock depending on having or not cancer. The prognosis, whether short or long term, was the same in both groups, advocating for not limiting admission to the ICU to cancer patients with septic shock.
I found the topic of great interest. The manuscript is very well written and concise, methods are well defined and the results are clearly presented.
Find below some questions and suggestions:
TITLE:
I suggest to change the title in order to better describe the objective of this work. For example “Comparison of short- and long term mortality in patients with or without cancer admitted to the ICU for septic shock: a retrospective observational study” or “Comparison of mortality between cancer patients and non-cancer patients admitted to the ICU for septic shock: a retrospective observational study”
ABSTRACT:
Introduction: Authors should be more explicit on the objective of this study that was to “compare the prognosis of cancer and non-cancer patients” instead of “study the updated outcomes”.
Methods: Authors should say that they looked at short- and long-term mortality with ICU and in-hospital mortality and 1 year survival.
Results: Authors omitted to state that although overall cancer status was not independently associated with in-hospital mortality, solid tumors were independently associated with in-hospital mortality
INTRODUCTION:
Authors should emphasize in the last sentence, the main objective of this study that was to compare the short and long term prognosis of patients with and without cancer admitted to the ICU for septic shock.
METHODS:
Why did you used KNAUS instead of PS ECOG to measure autonomy?
Line 107 : Maybe relace “dialysis » by «renal replacement therapy”
Please could you be more specific on which clinically relevant variables were kept in the model whatever was the association with the outcome in the univariate analysis?
Maybe you should state that you performed 2 MV models, one for ICU-mortality and 1 for in-hospital mortality.
RESULTS:
I suggest to raise the information about the % of patients with and without cancer above. The sentences (lines 154-158): “The 210 included patients were composed of 89 non-cancer patients (42.4%) and 121 cancer patients … respectively. “ should be pulled-up at the beginning of the results (after the 2nd sentence).
Do you know were came the patients from? I would be interested to know the % of patients admitted from the ED, the pre-hospital medical service and the wards?
Maybe I would have done 2 different tables:
- 1 with baseline characteristics, sites of infection, laboratory findings, treatment initiated before and during ICU
- 1 with LOS, short term and long-term outcomes
Because outcomes are split in the 2 existing tables.
The other solution is to remove the long term mortality from Table 1 because this information is given by the kaplan meier curve.
I am not sure to understand what did you do in table 3. This was an MV analysis to measure associations between variables and having a cancer? I do not understand the purpose of this analysis.
In the MV of factors associated with mortality, why some baseline characteristics such as Charlson and Knaus are not in the model. As the univariate analysis is not shown, it is difficult to know which variables were put in the model on the basis of the association with the outcome in the univariate model or because they were clinically relevant.
Author Response
Thank you for giving me the opportunity to review this manuscript. Authors compared the prognosis of patients admitted to the ICU with septic shock depending on having or not cancer. The prognosis, whether short or long term, was the same in both groups, advocating for not limiting admission to the ICU to cancer patients with septic shock.
I found the topic of great interest. The manuscript is very well written and concise, methods are well defined and the results are clearly presented.
Thank you immensely for your feedback. It is, indeed, a topic of great interest, and at the crossroads of several specialties (emergency medicine, critical care medicine and oncology-hematology).
Find below some questions and suggestions:
TITLE:
I suggest to change the title in order to better describe the objective of this work. For example “Comparison of short- and long term mortality in patients with or without cancer admitted to the ICU for septic shock: a retrospective observational study” or “Comparison of mortality between cancer patients and non-cancer patients admitted to the ICU for septic shock: a retrospective observational study”
We have amended the title accordingly by adopting the first of your suggestions.
ABSTRACT:
Introduction: Authors should be more explicit on the objective of this study that was to “compare the prognosis of cancer and non-cancer patients” instead of “study the updated outcomes”.
We have changed this in the abstract accordingly.
Methods: Authors should say that they looked at short- and long-term mortality with ICU and in-hospital mortality and 1 year survival.
We added this point the Methods sections.
Results: Authors omitted to state that although overall cancer status was not independently associated with in-hospital mortality, solid tumors were independently associated with in-hospital mortality
We corrected this point accordingly in the Results section.
INTRODUCTION:
Authors should emphasize in the last sentence, the main objective of this study that was to compare the short- and long-term prognosis of patients with and without cancer admitted to the ICU for septic shock.
We clarified that point by adding a sentence at the end of the introduction about the main objective of the study.
METHODS:
Why did you used KNAUS instead of PS ECOG to measure autonomy?
These two scores are quite similar, we chose to use the KNAUS score as it is more commonly used locally.
Line 107 : Maybe relace “dialysis » by «renal replacement therapy”
The terms were replaced accordingly.
Please could you be more specific on which clinically relevant variables were kept in the model whatever was the association with the outcome in the univariate analysis?
The clinically relevant variables kept in the model for ICU-mortality and in-hospital mortality, despite their non-significance, were age, on-going medical history of cancer, nosocomial infection and bacteremia.
Maybe you should state that you performed 2 MV models, one for ICU-mortality and 1 for in-hospital mortality.
This information was added in the Statistical Analysis section of the R1 manuscript.
RESULTS:
I suggest to raise the information about the % of patients with and without cancer above. The sentences (lines 154-158): “The 210 included patients were composed of 89 non-cancer patients (42.4%) and 121 cancer patients … respectively. “ should be pulled-up at the beginning of the results (after the 2nd sentence).
We modified our manuscript accordingly.
Do you know were came the patients from? I would be interested to know the % of patients admitted from the ED, the pre-hospital medical service and the wards?
In total, 144 (65.2%) patients were admitted to the ICU directly from the ward and 77 (34.8%) patients were admitted to the ICU from the ED. There was no direct admission to ICU directly from the pre-hospital medical service.
Maybe I would have done 2 different tables:
- 1 with baseline characteristics, sites of infection, laboratory findings, treatment initiated before and during ICU
- 1 with LOS, short term and long-term outcomes
We understand the Reviewer's remark, however the choice to split these two tables is voluntary, although it divided the outcome data. Our choice was guided by the fact that these tables were build with two different sample sizes (table 1 dealing with patients with history and long-term outcome and table 2 dealing with hospital stays, studying each stay in the ICU for septic shock and detailing the clinical characteristics, the management of septic shock and the in-hospital outcome). Therefore, we chose to keep the table split as it is for more clarity, as to not confuse ‘patient’ and ‘hospital stay’. However, we can modify the tables accordingly if the Editor or the Reviewers find it necessary.
Because outcomes are split in the 2 existing tables.
The other solution is to remove the long-term mortality from Table 1 because this information is given by the Kaplan meier curve.
I am not sure to understand what did you do in table 3. This was an MV analysis to measure associations between variables and having a cancer? I do not understand the purpose of this analysis.
The purpose of the table and this multivariate analysis was to visualize whether there were differences in terms of management, severity and outcome of the septic shock according to cancer status. We found no difference, except for more transfusion in the cancer group. This table analyzes the associations between variables and the presence of cancer, as pointed by the Reviewer, it is not of particular interest and has been put in supplemental data.
In the MV of factors associated with mortality, why some baseline characteristics such as Charlson and Knaus are not in the model. As the univariate analysis is not shown, it is difficult to know which variables were put in the model on the basis of the association with the outcome in the univariate model or because they were clinically relevant.
There were no significant association between Charlson score or Knaus score and ICU mortality. Concerning in-hospital mortality, only the Charlson score was significantly associated. However, we chose not to include either of these two scores in the model because they are not independent of the cancer disease of interest here.
Reviewer 2 Report
Le Borgne P et al. conducted a monocentric, retrospective cohort study on 210 patients admitted to ICU for septic shock, and compared patients with versus without cancer. When comparing ICU stays 28 of patients with versus without cancer (n=148 vs. n=91 stays, respectively), they found no significant difference in ICU and in-hospital mortality. Multivariate analysis showed that renal replacement therapy requirement, DIC, and mechanical ventilation were associated with ICU mortality, whereas malignancy, hematological or solid tumors, were not. In the same way, mortality was not significantly different at 365-day follow-up between the 2 groups.
From these data, the Authors suggest that malignancies should no longer be considered a barrier to ICU admission.
General comment: This paper is interesting, complete, and exhaustive in the description and analysis of the data. The conclusion is likely true, reflecting a different approach with new drugs, and improved outcomes for cancer patients.
1) the 1-year survival figure was similar in the 2 groups, and these figures indicate that the 1-year prognosis in the cancer group was not affected by the underlying neoplastic disease. In other words, the group that entered the ICU with cancer was in disease remission. This aspect must be underlined in the Materials Methods and the Discussion because it seems to be the real discriminating element for the access of the neoplastic patient in the ICU.
2) The group with cancer at entry had significant thrombocytopenia and leukopenia compared to the control group. Were these factors considered in the multivariate analysis? What does DIC stand for? For the presence of thrombocytopenia or the worsening of thrombocytopenia, e.g. due to Gram-negative sepsis?
Author Response
Comments and Suggestions for Authors
Le Borgne P et al. conducted a monocentric, retrospective cohort study on 210 patients admitted to ICU for septic shock, and compared patients with versus without cancer. When comparing ICU stays 28 of patients with versus without cancer (n=148 vs. n=91 stays, respectively), they found no significant difference in ICU and in-hospital mortality. Multivariate analysis showed that renal replacement therapy requirement, DIC, and mechanical ventilation were associated with ICU mortality, whereas malignancy, hematological or solid tumors, were not. In the same way, mortality was not significantly different at 365-day follow-up between the 2 groups.
From these data, the Authors suggest that malignancies should no longer be considered a barrier to ICU admission.
General comment: This paper is interesting, complete, and exhaustive in the description and analysis of the data. The conclusion is likely true, reflecting a different approach with new drugs, and improved outcomes for cancer patients.
Thank you for your feedback. Our conclusion indeed reflects different therapeutic approach, and improved outcomes for cancer patients. We are glad that this message is clear to the Reviewer.
1) the 1-year survival figure was similar in the 2 groups, and these figures indicate that the 1-year prognosis in the cancer group was not affected by the underlying neoplastic disease. In other words, the group that entered the ICU with cancer was in disease remission. This aspect must be underlined in the Materials Methods and the Discussion because it seems to be the real discriminating element for the access of the neoplastic patient in the ICU.
Thank you for this comment. We underlined this aspect by adding a clarifying sentence in the Discussion section.
2) The group with cancer at entry had significant thrombocytopenia and leukopenia compared to the control group. Were these factors considered in the multivariate analysis? What does DIC stand for? For the presence of thrombocytopenia or the worsening of thrombocytopenia, e.g. due to Gram-negative sepsis?
These factors (thrombocytopenia and leukopenia) were not considered in the multivariate analysis because they were not associated with mortality.
DIC is the known abbreviation for ‘disseminated intravascular coagulation’, a major prognostic factor in septic shock.
We included DIC in the multivariate analysis. Also, DIC scores included platelet measurement.
We did not consider the association between thrombocytopenia and Gram-negative bacteria infection, as it would have been limited by the sample size of our population and it was not the topic of question in this study.